# Investigation of AlGaN/GaN Heterostructures Grown on Sputtered AlN Templates with Different Nucleation Layers

**DOI:** 10.3390/ma12244050

**Published:** 2019-12-05

**Authors:** Chuan-Yang Liu, Ya-Chao Zhang, Sheng-Rui Xu, Li Jiang, Jin-Cheng Zhang, Yue Hao

**Affiliations:** 1State Key Discipline Laboratory of Wide Band Gap Semiconductor Technology, School of Microelectronics, Xidian University, Xi’an 710071, China; lcy2010@mail.usts.edu.cn (C.-Y.L.); shengruixidian@126.com (S.-R.X.); jchzhang@xidian.edu.cn (J.-C.Z.); yhao@xidian.edu.cn (Y.H.); 2School of Electronic and Information Engineering, Suzhou University of Science and Technology, Suzhou 215009, China; jiangli@mail.usts.edu.cn

**Keywords:** AlGaN/GaN heterostructure, sputtered AlN template, dislocation, stress relief effect, 81.05.Ea

## Abstract

In this work, a sputtered AlN template is employed to grow high-quality AlGaN/GaN heterostructures, and the effects of AlN nucleation layer growth conditions on the structural and electrical properties of heterostructures are investigated in detail. The optimal growth condition is obtained with composited AlN nucleation layers grown on a sputtered AlN template, resulting in the smooth surface morphology and superior transport properties of the heterostructures. Moreover, high crystal quality GaN material with low dislocation density has been achieved under the optimal condition. The dislocation propagation mechanism, stress relief effect in the GaN grown on sputtered AlN, and metal organic chemical vapor deposition AlN nucleation layers are revealed based on the test results. The results in this work demonstrate the great potential of AlGaN/GaN heterostructures grown on sputtered AlN and composited AlN nucleation layers for microelectronic applications.

## 1. Introduction

Group III nitride semiconductors have been demonstrated as promising candidates for optoelectronic and microelectronic applications due to their excellent material properties such as high absorption efficiency, high breakdown field, high thermal conductivity, and high saturated electron drift velocity [1,2,3,4,5,6]. Specially, GaN-based heterostructures possess enormous potential for fabricating high-power and high-frequency devices due to the high density and high mobility of two-dimensional electron gas (2DEG) formed by the polarization effect [7,8]. The power-gain cutoff frequency and output power density of the most advanced AlGaN/GaN high-electron-mobility transistors (HEMTs) have already exceeded 300 GHz and 30 W/mm [9,10].

GaN-based heterostructures are commonly grown on foreign substrates (SiC, Si, sapphire, etc.) due to the fact that larger diameter free-standing substrates were commercially unavailable up to now. Comparatively, the sapphire substrate is preferred due to its stability at high temperatures, its low cost, and its mature growth technology. However, the large lattice mismatch and thermal expansion coefficients mismatch between the sapphire substrate and the GaN epilayer lead to high dislocation densities and large stress in the growth material. Such issues have motivated the use of various nucleation layers (NL) in the GaN epitaxy on sapphire substrates. Among them, the sputtered AlN template on a sapphire substrate has attracted significant attention because the sputtering process supports a high throughput, is largely scalable, is inexpensive, and has been shown to give rise to reduced dislocation densities in the GaN grown thereon [11,12,13]. In terms of device performance, sputtered AlN films have been demonstrated to provide notable improvements in terms of increased light output, lower forward turn on voltage, reduced reverse leakage current, and improved reliability for visible and ultraviolet-light-emitting diodes [14,15].

However, there are few reports on the effects of a sputtered AlN template on the structural and transport properties of AlGaN/GaN heterostructures. Moreover, the differences in growth mechanisms of GaN grown on a sputtered AlN template and the metal organic chemical vapor deposition (MOCVD) NL were not fully investigated in the previous studies.

In this work, AlGaN/GaN heterostructures based on a sputtered AlN template and different MOCVD AlN NLs have been grown and investigated. Specially, the effect of different AlN NLs on the structural and transport properties of the heterostructures are discussed in detail. Moreover, high crystal quality GaN material with low dislocation density are achieved under the optimal growth conditions, and the dislocation propagation mechanism and stress relief effect are revealed.

## 2. Experimental Methods

Three AlGaN/GaN heterostructure samples (samples A–C) were grown on c-plane sapphire substrates, and the schematic structural diagrams are shown in Figure 1. The sapphire substrates employed in our experiment were single-side polished, and the size and thickness of the substrates were 4 inches and 650 μm, respectively. Firstly, a thin sputtered AlN layer was deposited on the sapphire substrates using a magnetron radio frequency reactive sputtering method with an aluminum (Al) target and nitrogen (N_2_)-argon (Ar) gas mixture. The purities of the Al target, N_2,_ and Ar were 99.999%, 99.995%, and 99.995%, respectively, and the background vacuum of the sputtering chamber was 10^−7^ Pa. The total gas flow rate during the sputtering process was maintained at 210 sccm while the proportion of the reactive N_2_ gas was fixed at 85%. The sputtering temperature and pressure were 0.4 Pa and 650 °C, respectively. The growth rate of the sputtered AlN was 15 nm/min and the growing time was 100 s, with a total thickness of 25 nm. Next, the sputtered AlN/sapphire substrates were transferred to a homemade MOCVD system with a showerhead reactor form to grow the heterostructures. Trimethyl-aluminum (TMAl), trimethyl-gallium (TMGa), and NH_3_ were used as the precursors for Al, Ga and *N*, respectively, and H_2_ was used as the metal organic source carrier gas. For sample A, at first, the sputtered AlN/sapphire substrate was baked for 300 s in H_2_ atmosphere at 900 °C. Next, a 1.6 μm GaN buffer layer was grown at 1080 °C. Finally, a 1 nm AlN interlayer, 25 nm AlGaN barrier layer with Al component of 0.23, and a 1 nm GaN cap layer were subsequently grown. For samples B and C, MOCVD AlN NL was inserted between the sputtered AlN template and the GaN buffer layer. A 160 nm high temperature (HT) AlN NL was employed for sample B, while a 40 nm low temperature (LT) and a 160 nm HT AlN NL were adopted for sample C. The growth temperatures for LT AlN and HT AlN NL were 900 °C and 1200 °C, respectively. Moreover, four additional samples (samples D–G) were also prepared. The growth conditions and parameters of samples D–F were exactly identical to those of samples A–C except for the absence of the AlGaN/GaN heterostructure, and the growth conditions and structures of sample G were exactly identical to that of sample C, except for the absence of the sputtered AlN layer. The growth temperature and thickness of each layer in samples A–G are summarized in Table 1.

The surface morphologies of the as-grown samples were characterized by a Bruker Icon Atomic Force Microscope (AFM), and the surface roughness was obtained by extracting the root mean square (RMS) values from a 2 × 2 μm^2^ scan area. The crystal quality of the obtained AlGaN/GaN heterostructures was evaluated using high-resolution X-ray diffraction (HRXRD) and cathode luminescence (CL) measurements. Both the ω-scan rocking curves from the symmetrical (002) and asymmetrical (102) reflections were measured with HRXRD, and the dislocation densities were estimated using the full width at half maximum (FWHM) of the rocking curves. Photoluminescence (PL) was also used to investigate the crystal quality of the GaN buffer layer. Raman spectroscopy was used to determine the strain state of the GaN epitaxial layers. Moreover, the electrical properties of the as-grown heterostructures were investigated by room temperature Hall-effect measurements in the van der Pauw configuration.

## 3. Results and Discussion

The surface morphology of the four AlGaN/GaN heterostructure samples were estimated by AFM in tapping mode, and the results in a 2 × 2 μm^2^ area scan are shown in Figure 2a. For all four samples, a clear atom step with some small pits can be observed, which is typical for group III nitride materials [16]. In addition, obvious fluctuations arise at the surfaces of samples A and G, indicating inferior homogeneity. For sample B, the height of the fluctuation, as well as the density of the small pits, are significantly reduced, suggesting improvement in the material quality and homogeneity. For Sample C, the surface is smooth and uniform, and no obvious fluctuation and small pits can be observed. The variation in surface morphology of the four samples is also reflected in the root mean square (RMS) roughness. The RMS roughness of sample A and G are 0.295 and 0.444 nm, respectively, and the value decreases to 0.181 and 0.109 nm, respectively, for samples B and C. We attribute the differences in the surface morphology to the variation of NLs in the four samples, and the optimized growth condition is the composited structure of LT and HT AlN NL deposited on the sputtered AlN template. In order to further investigate the effect of the MOCVD AlN NL on the surface morphology of the AlGaN/GaN heterostructures, three additional epitaxial samples named as samples D, E, and F were also prepared. As shown in Figure 1, the growth conditions of the additional samples are the same as those of samples A, B, and C, except for the absence of the AlGaN/GaN heterostructures. Figure 2a also shows the surface AFM images of samples D–F, and the corresponding 3D images are illustrated in Figure 2b. It can be observed that the surface morphology differs significantly from samples D, E, and F. For sample D, a high density of islands can be observed on the surface, indicating the polycrystal phase of the sputtered AlN. The surface RMS roughness of sample D is as high as 0.510 nm. For sample E with HT MOCVD AlN deposited on the sputtered AlN, the islands disappear and the surface shows a monocrystal phase, while the RMS roughness decreases to 0.361 nm. In addition, a mass of little pits arises, which is related to the dislocations. For Sample F, the surface shows clear atom steps and no pits can be observed, and the RMS roughness is as low as 0.131 nm. These results are consistent with the results in Figure 2a, demonstrating that the employment of a MOCVD AlN layer is beneficial for improving the surface morphology of the NL and the subsequent AlGaN/GaN heterostructures. Moreover, it is revealed that the deposition of composited LT and HT AlN on the sputtered AlN template is the optimized growth condition.

A HRXRD measurement was employed to investigate the crystal quality of the heterostructure samples with different NLs. Figure 3 displays the rocking curves from the (002) and (102) planes of samples A–C, and the normalized results are shown in the inset. Sample A presents the lowest full width at half maximum (FWHM) of 99 arcsec from (002) plane, and the values of samples B and C slightly increase to 153 and 125 arcsec. In the results from (102) plane, sample A shows the maximal FWHM of 667 arcsec, and the values of samples B and C significantly decrease to 367 and 337 arcsec. It is well-known that the FWHM values of (002) planes correspond to screw dislocation, and the FWHM values of (102) planes correspond to edge dislocation, respectively [17,18]. The densities of the dislocations can be calculated by [19,20]
$$D_{screw}=\frac{\beta_{\left(002\right)}^{2}}{4.36b_{screw}^{2}},\;D_{edge}=\frac{\beta_{\left(102\right)}^{2}}{4.36b_{edge}^{2}}$$ where *D_screw_* is the screw-type dislocation density, *D_edge_* is the edge-type dislocation density, *β* is the FWHM value measured by HRXRD rocking curves, and *b* is the burgers vector length (*b_screw_* = 0.5185 nm; *b_edge_* = 0.3189 nm). The screw dislocation densities of samples A, B, and C are 1.97 × 10^7^, 4.69 × 10^7^, and 3.13 × 10^7^ cm^−2^, and the corresponding edge dislocation densities are 2.42 × 10^9^, 7.14 × 10^8^, and 6.02 × 10^8^ cm^−2^, as shown in Table 2. The edge dislocation densities are more than one order higher than that of the screw dislocation. Previous reports have demonstrated that the vertical electric leakage of the device is related to the screw dislocation, while the edge dislocation significantly influences the horizontal transport properties of the carriers [21,22]. For samples B and C, the edge dislocation density, as well as the total dislocation density are significantly lower than those of sample A, and sample C with the LT and HT AlN layers shows the best result. In order to further confirm the results of HRXRD, cathode luminescence (CL) mapping was completed for samples A, B, C, and G, and the results are shown in Figure 4. The dark spots, which are associated with the nonradiative nature of threading dislocation (TD) in GaN material [23], are clearly observed in the CL mapping images of the four samples. The TD density is estimated to be 1.80 × 10^9^, 5.42 × 10^8^, 4.27 × 10^8^, and 1.75 × 10^9^ cm^−2^ for samples A, B, C, and G, respectively. The CL results show the same trend as the HRXRD results, indicating the highest TD density in sample A and the lowest density in sample C. In addition, the TD density estimated from CL mapping is slightly lower than the value calculated based on the HRXRD results. This phenomenon resulted from the difference in the test depth between the CL and HRXRD measurements.

The dislocation suppression effects of the MOCVD AlN layers are illustrated in Figure 5. According to the model proposed by Fischer [24], the critical relaxation thickness of the epilayers can be calculated as
$$d_{cr}\approx b/2\varepsilon$$ where *b* is the magnitude of the Burgers vector and *ε* is the in-plane biaxial strain. The calculated critical thicknesses of AlN on sapphire and GaN on AlN are 1.03 and 6.29 nm, respectively, which are far below the real thickness in our experiment. Therefore, a mass of misfit dislocation will arise at the heteroepitaxy interfaces and further propagate as TD along the growth orientation. For the GaN grown directly on the sputtered AlN template, the TD extends to the GaN buffer without any annihilation effect. Thus, the TD density of sample A is high. However, for samples B and C with HT AlN NL, the deposition atoms can get more energy, and the lateral transfer effect, as well as the lateral growth rate of material, are enhanced. As a result, the lattice planes in the HT AlN NL will twist due to the strain effect, while the dislocations will bend and then annihilate when contacting with each other. Therefore, the TD densities in samples B and C are obviously lower than the result of sample A. Moreover, for sample C, the LT AlN NL possesses high-density nucleation islands, and the TD also bends when reaching the inclined sidewalls of the islands. Therefore, the sample C has the lowest TD density. These results indicate that the use of LT and HT AlN NLs on the sputtered AlN template is beneficial for improving the material quality of the subsequent GaN by suppressing the edge dislocation density.

In order to investigate the influence of the different NLs on the optical properties of materials, a PL measurement was performed for samples A–C using the 325 nm line of an Ar+ laser as the excitation source. As shown in Figure 6, all three AlGaN/GaN heterostructure samples show a clear near band-edge emission peak at around 361 nm without obvious yellow and blue luminescence, indicating the low-point defect concentration in the three samples. However, obvious differences in the intensity of the near band-edge emission peaks of the three samples can be observed. Sample C shows the highest peak intensity, which is 5.41 and 2.76 times higher than that of samples A and B, respectively. This phenomenon is consistent with the HRXRD results, demonstrating that the employment of composited LT and HT MOCVD AlN NL is beneficial to improve the crystal quality of GaN.

Raman measurement was employed to detect the stress state of the GaN materials in the three samples, and the results are shown in Figure 7. All three samples show the phonon mode of E_2_ (high), A_1_ (LO), and E_1_ (LO). The position of the phonon mode of E_2_ (high) is usually used to evaluate the stress state in GaN material, and the mode of strain-free GaN has been revealed at around 567.5 cm^−1^ [25,26]. The E_2_ (high) mode of samples A, B, and C locate at 570.79 cm^−1^, 571.86 cm^−1^, and 571.32 cm^−1^, respectively, indicating that all the three GaN samples suffer from compressive stress. In addition, the intensity of the stress can be calculated as 0.783 GPa, 1.037 GPa, and 0.910 GPa for samples A, B and C, respectively. According to the HRXRD results, the GaN sample grown directly on the sputtered AlN template (sample A) possessed the highest dislocations density, and thus, the compressive stress was released by the formation of misfit dislocations and TDs. However, sample C has the lowest dislocation density, while the stress intensity is lower than that of sample B. This phenomenon can be explained as follows. High-density nucleation islands with different crystal orientations and crystal structures (sphalerite and wurtzite) can be formed in the course of LT AlN layer growth. These nucleation islands can infiltrate sapphire substrates to partially release the stress caused by lattice mismatch between substrate and epitaxial material [27,28]. Therefore, the residual compressive stress in the subsequent GaN buffer layer is reduced. For sample B, the lack of infiltration of the LT AlN NL on the sapphire substrate leads to large compressive stress in HT AlN NL and increases the compressive stress in the GaN buffer layer.

In order to further investigate the transport properties of the four AlGaN/GaN heterostructure samples, a Hall effect measurement was taken, and the results of mobility, 2DEG density, and sheet resistance are shown in Table 3. On one hand, the 2DEG sheet density shows a decrease trend from the order of samples B, C, and A, which is consistent with the stress intensity as shown in the Raman results. The compressive stress intensity reflects that the lattice constant of GaN monotonously decreases in the order of samples B, C, and A, resulting in the monotonously decreases in piezoelectric polarization intensity and the 2DEG density. On the other hand, the mobility increases monotonously in the order of samples G, A, B, and C, which can be explained by two aspects. Firstly, as the AFM results show, the surface morphology can partly reflect the flatness of the AlGaN/GaN interface. Therefore, it can be inferred that the interface roughness scattering intensity decreases monotonously in the order of samples G, A, B, and C. Secondly, the results of the HRXRD and the CL demonstrate that the dislocation density decreases monotonously in the order of samples G, A, B, and C, leading to a decrease in dislocation scattering. As a result, the 2DEG mobility of sample C is higher than that of the other samples.

## 4. Conclusions

In summary, high-quality AlGaN/GaN heterostructures were grown on sputtered AlN templates and the influences of MOCVD AlN NL growth conditions on the structural and electrical properties of the heterostructures were investigated in detail. The lowest surface RMS, smallest dislocation density, and highest 2DEG mobility are achieved in the case that AlGaN/GaN heterostructure is grown on composited MOCVD AlN NL and sputtered AlN template. Moreover, the dislocation propagation mechanism and stress relief effect in the GaN grown on sputtered AlN and metal organic chemical vapor deposition AlN nucleation layers were revealed. The results not only demonstrate the great potential of AlGaN/GaN heterostructures grown on composited AlN NL for microelectronic applications, but they also provide an instructive way to further research on the growth of high-quality group III nitride materials.

## Figures and Tables

**Figure 1 materials-12-04050-f001:**
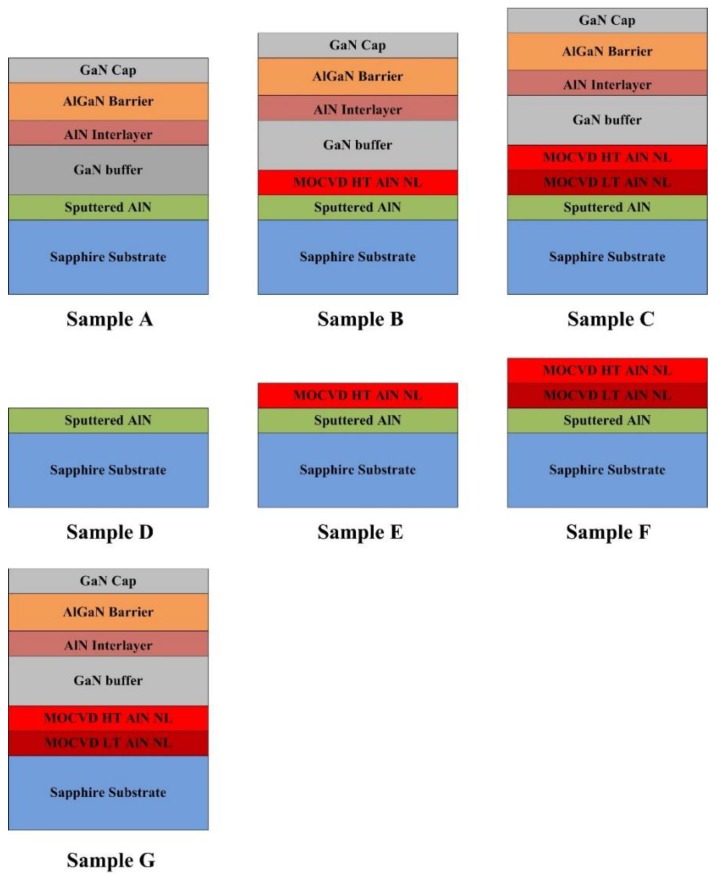
Schematic cross-structural diagram (not to scale) of this work.

**Figure 2 materials-12-04050-f002:**
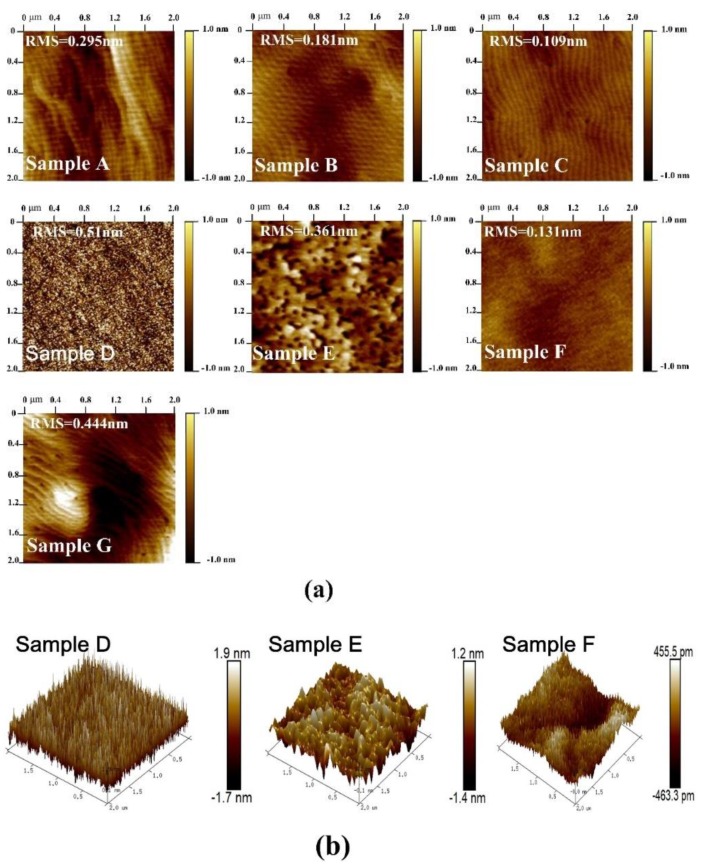
AFM images of all samples: (**a**) 2D AFM images of samples A–G and (**b**) 3D AFM images of samples D–F.

**Figure 3 materials-12-04050-f003:**
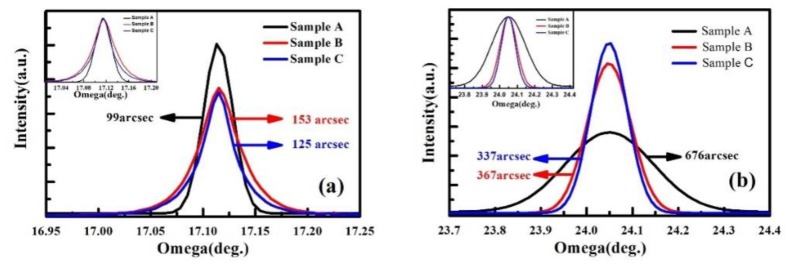
High-resolution X-ray diffraction (HRXRD) rocking curves of (**a**) (002) plane and (**b**) (102) plane scans for samples A–C. The insets show the normalized results.

**Figure 4 materials-12-04050-f004:**
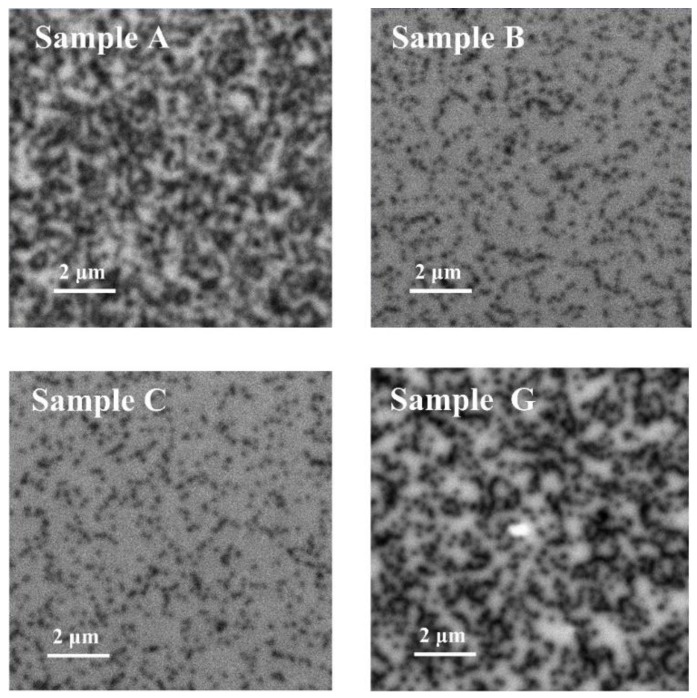
Cathode luminescence (CL) mapping images of samples A, B, C, and G.

**Figure 5 materials-12-04050-f005:**
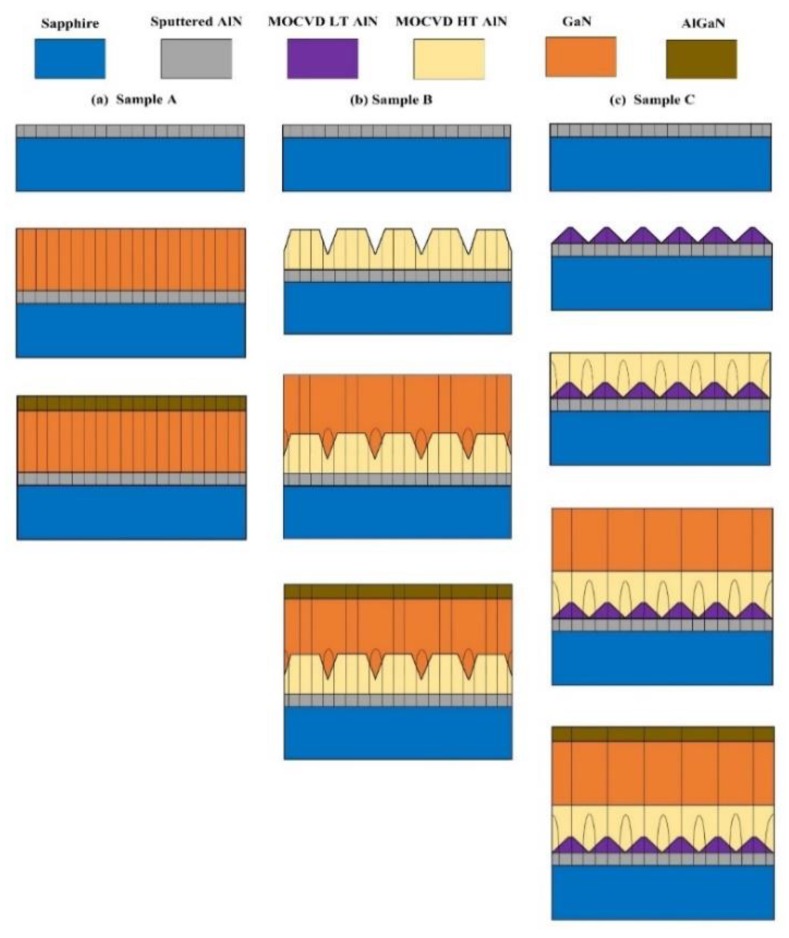
Schematic dislocation evolution diagram in samples A, B, and C.

**Figure 6 materials-12-04050-f006:**
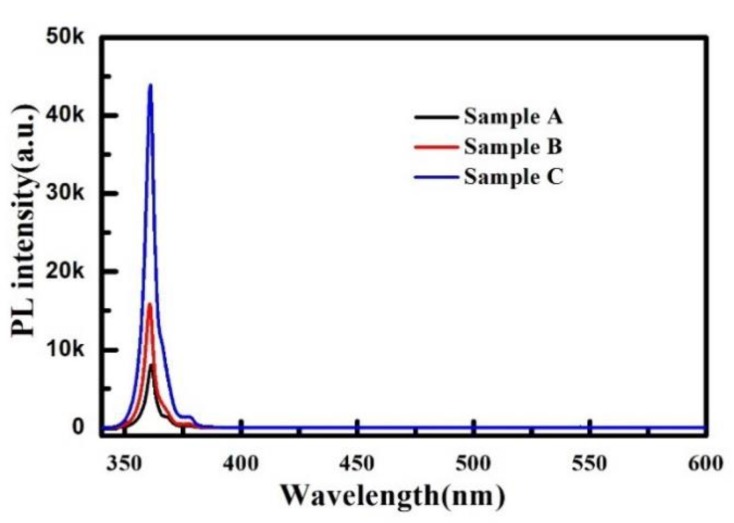
Photo luminescence (PL) spectra for samples A–C with a wavelength of 340 to 600 nm at room temperature.

**Figure 7 materials-12-04050-f007:**
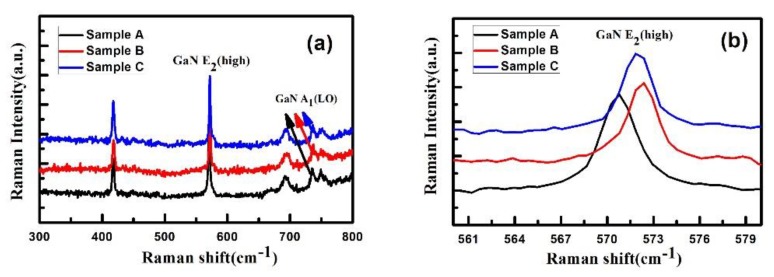
Raman spectra for samples A–C at room temperature. (**a**) Raman spectra from 300 cm^−1^ to 800 cm^−1^ and (**b**) Raman spectra from 560 cm^−1^ to 582 cm^−1^ near GaN E_2_ (High) mode.

**Table 1 materials-12-04050-t001:** Layer parameters of samples A–F.

Sample	Sputtered AlN	LT AlN	HT AlN	GaN Buffer	AlN Interlayer	AlGaN Barrier	GaN Cap
#A	25 nm	×	×	1.6 μm	1 nm	25 nm	1 nm
	650 °C			1080 °C	1080 °C	1080 °C	1080 °C
#B	25 nm	×	160 nm	1.6 μm	1 nm	25 nm	1 nm
	650 °C		1200 °C	1080 °C	1080 °C	1080 °C	1080 °C
#C	25 nm	40 nm	160 nm	1.6 μm	1 nm	25 nm	1 nm
	650 °C	900 °C	1200 °C	1080 °C	1080 °C	1080 °C	1080 °C
#D	25 nm	×	×	×	×	×	×
	650 °C						
#E	25 nm	×	160 nm	×	×	×	×
	650 °C		1200 °C				
#F	25 nm	40 nm	160 nm	×	×	×	×
	650 °C	900 °C	1200 °C				
#G	×	40 nm	160 nm	1.6 μm	1 nm	25 nm	1 nm
		900 °C	1200 °C	1080 °C	1080 °C	1080 °C	1080 °C

**Table 2 materials-12-04050-t002:** Dislocation density of samples A–C (Unit: cm^−2^).

Sample	A	B	C
MOCVD AlN NL	without	HT	LT and HT
$D_{screw}$	1.97 × 10^7^	4.69 × 10^7^	3.13 × 10^7^
$D_{edge}$	2.42 × 10^9^	7.14 × 10^8^	6.02 × 10^8^
$D_{\mathrm{total}}$	2.44 × 10^9^	7.61 × 10^8^	6.33 × 10^8^

**Table 3 materials-12-04050-t003:** Electrical properties of samples A, B, C and G

Sample	A	B	C	G
MOCVD AlN NL	without	HT	LT and HT	LT and HT without sputtered AlN
Sheet resistance (Ω/sq)	383.27	330.95	329.67	450.40
2DEG Mobility (cm^2^/V·s)	1901.97	1987.44	2050.70	1775.97
2DEG density (cm^−2^)	8.56 × 10^12^	9.49 × 10^12^	9.23 × 10^12^	7.80 × 10^12^

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
