# Peer review of "Investigation of AlGaN/GaN Heterostructures Grown on Sputtered AlN Templates with Different Nucleation Layers"

_materials, 2019, doi:10.3390/ma12244050_

Round 1

Reviewer 1 Report

It is a really interesting contribution presenting a new simple approach to improve the GaN crystal quality when grown on sapphire using different AlN grown layers. The contribution is interesting and the paper should be published after some improves.

The following aspects should be modified:

1.- Page 4: Fig.1, The dark colour of the AlGaN layer make the indication not “readable”

2.- Page 3: line 66-74, a table should be introduced where for sample A to F, each layer condition should be detailed:

AlN sputtered

LT AlN

HT AlN

….

….

….

#A

#B

#C

#D

#E

#F

3.-Page 3: line 63 acronym TMAl has not been explained before.

4..- Page 5, line 94, The surface morphology CAN NOT BE MEASURED!!!! Better to indicate “… heterostructure samples morphology is estimated by AFM….”

5.- Page 6, Fig.2: sample indentification should be inserted on each micrograph.

6.- Page 6, line 137, the text indicate that “…it is well known…”, references should be then cited.

7.- Page 8, Fig4, dislocation bending inside the MOCVD HT AlN is not explained. Why it occurs? Does the expansion coefficient play a role in the dislocation behaviour through a different strain state?

8.- Page 9, line 199, the referee has some discrepancies with the text. TD do NOT change significantly the stress state. Misfit dislocation are required for this….. Another explanation is required.

9.- The feeling of the referee is that the underneath AlN respect to the HT-AlN has TD that bends at the LT/HT interface making misfit dislocation at it interface. The HT-AlN is then partially relaxed, as observed by Raman. Indeed, HT increase the dislocation mobility and then stress make them bent. For this the Matthew and Blakeslee critical thickness should be attained. The author can calculate it. This should be added to the manuscript. TEM observations can also evidence it.

10.- Page 10, Fig.6, Inset indication and scales not visible. The size of the letters should be increased or the inset changed to a Fig.6b to increase size.

The point 9 indicate that the manuscript needs some strong modifications. The physics of the dislocation behaviour is NOT explained. At least a tentative explanation is required.

Author Response

Thank you very much for your email dated 13th November 2019, regarding the reviewers’ comments on our manuscript (#materials-632198 with revision) entitled “Investigation of AlGaN/GaN heterostructures grown on sputtered AlN template with different nucleation layers”.

Firstly, thanks very much for the reviewer’s recognition of our manuscript.

We are grateful for the patient corrections from the editor and reviewers, which have improved the quality and significance of our manuscript. According to the reviewers’ comments, we have made appropriate revisions as presented in the response letter.

Reviewer 2 Report

The article investigated the quality of AlGaN / GaN structures grown by the MOCVD method on a sapphire substrate with an AlN buffer layer deposited by sputtering,
using various intermediate layers.
Three types of buffer layers were investigated: a sputtered AlN buffer only, sputtered AlN + MOCVD HT-AlN, and sputtered AlN + MOCVD LT-AlN - MOCVD HT-AlN.
Surface morphology, Carrier density and mobility, and XRD rocking curve (002) and (102) were investigated.

The study has serious flaws and can not be published in the present form.

1) Introduction/Literature review
Most links in the Literature review [1-20] describe performance and applications of state-of-the-art GaN-based devices
and are not connected with the main paper topic - buffer layers for high-quality GaN/AlGaN heteroepitaxy.
Only five references [21-25] are devoted to AlGaN LEDs grown on sputter-deposited buffer layers, while references with 2DEG structures are missing.
It makes it hard to compare the results with the contemporary technology level.

2) Experimental methods description
No information is given on the sputter-deposited AlN template:
- what was the thickness of the sapphire substrate?
- what was the sputtering method used?
- what were the sputtering conditions - temperature, pressure, atmosphere composition, target composition and purity, deposition rate, etc.?
- What was the composition of the sputtered AlN layers? Level of oxygen contamination?
What was the type of MOCVD reactor used?

No comparison is given with AlGaN structures grown directly on the sapphire substrate, without sputter-deposited buffer.
It is not clear if this layer has any significant influence on the device quality, thus making the relevance of the study questionable.
From the results, it seems that the MOCVD-AlN buffer plays a major role in structure quality.

Dislocation density is given with 3-digit precision. 2DEG Sheet-resistance, Hall mobility, and 2DEG density are also given with physically meaningless precision. What was the real measurement precision?
Dislocation density was estimated indirectly using XRD data.
More direct studies like micro-CL or TEM should be used to discuss the dislocation density soundly.

4) Conclusions
The authors claim "very low dislocation density" achieved using optimal conditions; however, the dislocation density of order 10^9cm-2 seems to be high for state-of-the-art AlGaN/GaN structures.

Author Response

(The authors gave the same response as above.)

Round 2

Reviewer 2 Report

The authors carefully revised the manuscript and improved it significantly, addressing major points of the review.

Additional data comparing structures grown on bare sapphire and on an AlN sputtered layer were presented in the author's reply (Fig. 2).

However, it is highly recommended to add this data to the manuscript (including the scheme in Fig. 1 on the manuscript, AFM data to the fig. 2, CL mapping to fig. 4 and Hall data to table III).

After these corrections, the manuscript may be accepted for publication.

Author Response

Dear Reviewer:

Thank you very much for your email dated 27th November 2019, regarding the reviewers’ comments on our manuscript (#materials-632198 with revision) entitled “Investigation of AlGaN/GaN heterostructures grown on sputtered AlN template with different nucleation layers”.

Firstly, thanks very much for the reviewer’s recognition of our manuscript.

We are grateful for the patient corrections from the editor and reviewers, which have improved the quality and significance of our manuscript. According to the reviewers’ comments, we have made appropriate revisions as presented below.
